# Avoiding High Pressure Abdominal Closure of Congenital Abdominal Wall Defects—One Step Further to Improve Outcomes

**DOI:** 10.3390/children10081384

**Published:** 2023-08-14

**Authors:** Raluca-Alina Mocanu, Cătălin Cîrstoveanu, Mihaela Bizubac, Ionuț Fernando Secheli, Nicolae Sebastian Ionescu

**Affiliations:** 1Doctoral School of “Carol Davila” University of Medicine and Pharmacy, 020021 Bucharest, Romania; raluca-alina.mocanu@drd.umfcd.ro; 2Pediatric Surgery Department, ‘M.S. Curie’ Emergency Clinical Hospital for Children, 041451 Bucharest, Romania; tacclo@hotmail.com (I.F.S.); sebastian.ionescu@umfcd.ro (N.S.I.); 3Department of Neonatal Intensive Care, “Carol Davila” University of Medicine and Pharmacy, 020021 Bucharest, Romania; ana-mihaela.bizubac@drd.umfcd.ro; 4Neonatal Intensive Care Unit, “M.S. Curie” Emergency Clinical Hospital for Children, 041451 Bucharest, Romania; 5Department of Pediatric Surgery and Orthopedics, “Carol Davila” University of Medicine and Pharmacy, 020021 Bucharest, Romania; 6Romanian Academy of Medical Sciences, 030167 Bucharest, Romania; 7Romanian Academy of Scientists, 030167 Bucharest, Romania

**Keywords:** intraabdominal pressure, compartment syndrome, gastroschisis, omphalocele

## Abstract

The main goal of surgical treatment for gastroschisis and omphalocele is the reduction of viscera in the abdominal cavity and closure of the abdomen, but the challenge is to succeed without the detrimental effects of increased intraabdominal pressure. In this regard, we performed a retrospective study for all patients admitted for gastroschisis and omphalocele to the Neonatal Intensive Care Unit of ‘Marie Sklodowska Curie’ Emergency Clinical Hospital for Children, from January 2011 until June 2021. Our aim was to highlight the presence of postoperative abdominal compartment syndrome. We observed that six out of forty-seven patients developed clinical signs of abdominal compartment syndrome, five associated with primary closure and one with staged closure with a polyvinyl chloride patch. Following the results, we decided to implement the trans-bladder measurement of intraabdominal pressure to avoid closing the abdomen at pressures higher than 10 mmHg in order to prevent the development of abdominal compartment syndrome. We consider that there is still place for the improvement of congenital abdominal wall defects management and that the measurement of intraabdominal pressure might help us reach our goal.

## 1. Introduction

It is well-known that the surgical objective in gastroschisis and omphalocele is to integrate the viscera in the abdominal cavity and then to restore the abdominal wall [1]. But we must take into account that visceral–abdominal disproportion impairs repositioning viscera in the abdominal cavity and, in this regard, intra-abdominal pressure (IAP) represents the keystone for postoperative evolution of patients with congenital abdominal wall defects. 

Increased IAP has an impact on several organs and systems. The respiratory function is influenced by the elevation and compression of the diaphragm, causing decreased tidal volume, atelectasis, hypoxemia, and hypercarbia [2,3]. Due to compression on the mesenteric vessels, portal vein, and inferior vena cava, increased IAP leads to decreased venous return to the right heart and consequently decreased cardiac output. Compression of the heart reduces its compliance and decreases the stroke volume, leading to diminished cardiac output [4]. Owing to decreased cardiac output and compression on renal parenchyma and renal vessels, renal perfusion declines, leading to a reduced glomerular filtration rate [5,6]. Reduced mesenteric circulation, as a consequence of increased IAP, causes poor intestinal perfusion and ischemia and also diminishes lymphatic drainage with intestinal edema and bacterial translocation [7]. Cerebral venous return is impaired by increased central venous pressure due to increased intrathoracic pressure, causing hypoxia and edema manifesting as lethargy, irritability, somnolence, or unresponsiveness [2,8].

The World Society of the Abdominal Compartment Syndrome (WSACS) established different thresholds when defining intraabdominal hypertension (IAH) for adults and for children. According to the 2006 consensus, IAP ≥ 2 mmHg defines IAH [9]. Subsequently, 2013 WSACS consensus changes IAH definition to IAP > 10 mmHg, only for pediatric patients [10]. Furthermore, abdominal compartment syndrome (ACS) is defined differently for adults and for children. ACS is defined, for adults, as IAP > 20 mmHg associated with new organ dysfunction and, for children, as IAP > 10 mmHg associated with worsening or new organ dysfunction [9,10]. 

Due to the fluid-like behavior of the abdomen, we can apply Pascal’s law, which states that the force is uniformly transmitted in a confined fluid and we can measure the pressure in any point [11,12]. Therefore, there are various sites for the indirect measuring of IAP described in the literature, namely, bladder, stomach, rectum, uterus, or inferior vena cava [13]. An experimental animal study compared the direct measurement of IAP with piezoresistive probes placed in the upper and lower abdomen, and the indirect measurement of intravesical pressure (IVP) and intragastral pressure. The mean baseline pressure values were similar for the caudal peritoneal probe and IVP measurement [14]. Other studies comparing pressures measured via an indwelling urinary catheter and via an intraperitoneal dialysis catheter demonstrated that IVP represents an accurate delineation of IAP [15,16]. Another study, performed on patients undergoing laparoscopic cholecystectomy, revealed that urinary bladder pressure showed parallel increments to elevated IAP during intraperitoneal carbon dioxide insufflation and also a similar decrease of IVP during gradual deflation of the intraperitoneal cavity [17].

IVP represents a valid reflection of IAP, and WSACS consensus from 2013 recommends the indirect trans-bladder technique for the measurement of IAP [10]. 

The aim of the study was to highlight the occurrence of postoperative abdominal compartment syndrome in patients treated for gastroschisis and omphalocele, in the Neonatal Intensive Care Unit of “Marie Sklodowska Curie” Emergency Clinical Hospital for Children, over the last decade. Identifying the pitfall of postoperative abdominal compartment syndrome when performing abdominal closure without quantifying the IAP, we decided to start monitoring IVP in order to have an objective evaluation of the IAP. We intend to improve our outcomes by adjusting the surgical procedure based on IVP measurement in an attempt to avoid IAH.

## 2. Materials and Methods

We performed a retrospective study on 47 patients born with gastroschisis and omphalocele that were admitted from January 2011 to June 2021 to the Neonatal Intensive Care Department of “Marie Sklodowska Curie” Emergency Clinical Hospital for Children.

Data were extracted retrospectively from our hospital medical records. We developed a database that included the following parameters: gender, gestational age, birth weight, Apgar score, type of abdominal wall defect, associated congenital anomalies, type of surgical procedure performed for abdominal wall closure, additional surgical procedures, time from birth until surgical procedure, and presence of abdominal compartment syndrome. The identification data were not included in our database. Our main aim was to emphasize the presence of postoperative abdominal compartment syndrome for patients with gastroschisis and omphalocele.

Starting from July 2021, we introduced IVP measurement as a means to assess patients with congenital abdominal wall defects, and we began a prospective study with the purpose of reducing the incidence of postoperative abdominal compartment syndrome. We included in the study all the patients with congenital abdominal wall defects that were admitted to our Neonatal Intensive Care Unit. There were no exclusion criteria for this study. Up to this point, we included in our survey nine patients with gastroschisis and omphalocele.

We inserted a Foley catheter in the bladder and connected it to a pressure transducer that is usually used in our Neonatal Intensive Care Unit for invasive blood pressure monitoring. The patient was placed in the supine position, with no elevation of the head of the bed. Then, we introduced in the bladder, through the Foley catheter, 1 mL/kg of room-temperature sterile 0.9% NaCl solution. After 1 min, we read the value of IVP shown on the monitor as a substitute of central venous pressure in mmHg (Figure 1).

We created a database similar to the one for the retrospective study and, additionally, we included daily measurement of arterial pressure, heart rate, diuresis, and oxygen saturation and noted the value of Peak Inspiratory Pressure (PIP) for mechanically ventilated patients, the presence of intestinal transit, and the presence of edema.

Data were recorded using Microsoft Excel (Microsoft Corporation (2018). Microsoft Excel) and were analyzed using statistical software JASP (JASP Team (2022). JASP (Version 0.16.3) [Computer software])

Informed consent for surgery was obtained from the parents of all the subjects involved in the study. Operative technique was not standardized and was chosen by the lead surgeon based on their experience.

Since 2018, our informed consent has included a section dedicated to permission to participate in clinical teaching and to use their medical data for research studies, considering our institution is a teaching hospital. For the cases prior to 2018, the medical data collection was approved by the ethics committee of our hospital. 

The approval of the ethics committee was obtained both in terms of data collection and analysis and in terms of publishing the results (no. 11231/31 March 202).

## 3. Results

### 3.1. Retrospective Study Results

Between January 2011 and June 2021, forty-seven patients with gastroschisis and omphalocele were admitted to the Neonatal Intensive Care Department of our hospital, of whom 30 were diagnosed with gastroschisis and 17 with omphalocele. 

Demographics for each group are described in Table 1. We can observe that there are no significant differences between the analyzed parameters, with a mean birth weight of 2404 g (±507 g) for the gastroschisis group and 2681 g (±591 g) for the omphalocele group, a mean gestational age of 36 weeks for both groups, and a mean Apgar score of 7 (±1) for the gastroschisis group and 8 (±1) for the omphalocele group.

In the gastroschisis group, the number of male and female patients was equal: fifteen males and fifteen females, while in the omphalocele group it was a female preponderance: eleven female patients and six male patients. 

Additional congenital anomalies were identified in twenty-three of the gastroschisis patients, of whom seven had multiple associated anomalies (more than two anomalies in different organ systems) and sixteen had a single additional anomaly. Additional anomalies were found in fifteen omphalocele patients, of whom five had multiple associated anomalies and ten had a single additional anomaly. Figure 2 summarizes the distribution of associated anomalies related to the affected organ system.

There were no standardized criteria for electing the type of abdominal wall closure, the decision of the surgical procedure being based on the experience of the surgeon. Primary closure was chosen for 27 of the 30 patients with gastroschisis, in five cases using additional abdominal wall stretching and in one case using dissection to create skin flaps for closure of the abdomen. In the other six cases, an umbilical cord patch was used for primary closure. Two patients needed intestinal resection and anastomosis due to associated intestinal atresia. Staged closure was chosen in three cases using three different methods: using a polyvinyl chloride tailored bag, a silo bag, or a polyvinyl chloride patch (Table 2). 

For omphalocele, primary closure was chosen in 16 of the 17 cases. Delayed closure was performed in one case, after daily wound dressing for 28 days. Regarding additional procedures, abdominal wall stretching was used in one case of primary closure. For one patient with ileal atresia and Ladd band, intestinal resection with ileostomy and lysis of the Ladd band was performed, and one patient with patent omphaloenteric duct required resection of the omphaloenteric duct (Table 3).

On the grounds that gastroschisis requires emergency surgical intervention, we performed surgery shortly after the patients were admitted to the Neonatal Intensive Care Unit. Because patients are transferred to our hospital from different institutions, the time from birth to surgical procedure varied according to transport time. For gastroschisis, he median time from birth to surgical procedure was 6 h, with a standard deviation of 5.85 h. For omphalocele, the surgical procedure can be postponed since this congenital anomaly does not represent a surgical emergency if the omphalocele membrane is not ruptured. For omphalocele there was a median time of 48 h, with a minimum time from the delivery until surgical procedure of 7 h and a maximum time of 672 h for the patient that was managed 4 weeks with daily wound dressings before the abdominal wall closure was performed (Table 4).

Studying the medical files, we observed that 12.77% of the patients developed clinical signs of abdominal compartment syndrome. From the total of six patients with abdominal compartment syndrome, five patients with gastroschisis underwent primary closure of the abdominal wall and the sixth underwent staged closure using a polyvinyl chloride patch (Table 5).

For the patients included in the retrospective study, abdominal compartment syndrome was diagnosed based on clinical signs. Based on their medical charts, all six patients had tense abdomen upon palpation. Decreased urinary output and an increase in serum creatinine levels were used to diagnose renal failure. According to the Neonatal Acute Kidney Injury (AKI) classification of Kidney Disease Improving Global Outcomes (KDIGO) [18], there were three patients with stage 1 AKI, two patients with stage 2 AKI, and one patient with stage 3 AKI. Moreover, one of the patients developed tachycardia and hypotension and was diagnosed with acute heart failure.

Regarding medical management of the abdominal compartment syndrome, medical treatment with diuretics (Furosemide) was prescribed for all the patients. Four patients required inotropic support with Dopamine, two of them needed Dopamine and Adrenaline administration and, for the patient with acute heart failure, Dopamine, Adrenaline, and Dobutamine were administered.

Four of the six patients described with ACS needed subsequent surgical procedures. Due to abdominal distension and the absence of stools, one patient required surgical intervention, and a 4 cm segment of ileal stenosis was identified with intestinal distension proximal to this segment. The stenotic segment was resected and Mikulicz ileostomy was performed. Closure of the abdomen was not possible due to abdominal–visceral disproportion, and prosthetic material (Gore-Tex Dual Mesh) was used for abdominal wall closure. Two weeks later, intestinal anastomosis was performed to restore intestinal continuity and the abdominal wall was restored. One patient had jejunal stenosis, and segmental resection with end-to-end jejunal anastomosis was performed one month after the first procedure. Two patients required lysis of bowel adhesions and one of them had a covered ileal perforation; therefore, an ileostomy was performed. Gentle maneuvers for intestinal content evacuation were performed for the last three patients described. In that manner, the previously distended bowel loops were easily reintegrated in the abdominal cavity and closure of the abdominal wall without tension was possible.

### 3.2. Prospective Study Preliminary Results

Since we implemented the IVP measurement, nine patients with congenital abdominal wall defects were admitted in our hospital—five patients with gastroschisis (three males and two females) and four with omphalocele (two males and two females).

The mean birth weight was 2835 g, with a minimum weight of 2000 g and a maximum weight of 3900 g. The gestational age ranged between 37 weeks and 40 weeks, and the Apgar score ranged between 7 and 9 (Table 6).

Regarding associated congenital anomalies, four patients with gastroschisis and three patients with omphalocele had cardiac malformations: all seven were diagnosed by sonography with atrial septal defect and patent ductus arteriosus and one of them with aortic arch stenosis. At the time of surgery, two of the patients with omphalocele and all of the gastroschisis patients were diagnosed with intestinal malrotation; two of the patients with gastroschisis also had associated microcolon. Additionally, one patient with gastroschisis and one with omphalocele had associated renal ectopia. 

Two patients with gastroschisis underwent primary closure and three underwent staged closure. For staged closure, a modified Schuster procedure was performed using a suspended tailored polyvinyl chloride (PVC) bag with serial reduction while measuring IVP and later reintegration. In the omphalocele group, two patients had primary closure, one patient underwent partial reduction and ligation of the omphalocele sac with abdominal wall closure the next day, and another underwent staged closure with the Schuster procedure using a suspended tailored polyvinyl chloride bag and serial reduction (Table 7).

Additionally, for the patients that underwent the Schuster procedure, we measured the IVP at the time of the serial reductions and one or two hours after the procedure. In the cases where we performed staged closure, there were no complications related to the type of prosthetic material we used to create the bag for gradual reduction of the viscera.

We evaluated daily respiratory and cardiac function, measured diuresis, and observed intestinal transit. There were no clinical signs of developing abdominal compartment syndrome and the IVP values monitored were all below 10 mmHg. 

Normal values of diuresis were noted, with a minimal value of 1 mL/kg/h in one case for a day, and all the other recorded values of diuresis were above 2 mL/kg/h (Table 8).

Regarding cardiac function, Table 9 and Table 10 depict the value ranges of heart rate and mean arterial pressure (MAP). We reported values of heart rate and MAP between the normal ranges according to the reference values for the neonatal period. 

We monitored IVP in the Operating Room at the time of abdominal wall closure, then one or two hours after the procedure in the Neonatal Intensive Care Unit, and then daily until we reached a plateau. A descriptive analysis of the IVP values measured displays no value above 10 mmHg (Table 11).

Figure 3 is a graphical representation of the evolution of IVP values recorded postoperatively. With the red line, we marked the 10 mmHg threshold, as this is the value above which WSACS consensus defines IAH. Variations of pressure values were observed, especially for patients with staged closure—pressure increased after ligation of the PVC tailored bag and after definitive abdominal wall repair. For instance, Patient 6 underwent staged closure with the modified Schuster procedure with a PVC tailored bag, for a giant omphalocele (liver, spleen, small bowel, and colon protruding through the abdominal wall defect). When serial reduction of the viscera in the abdomen with ligation of the bag was performed, the IVP increased as shown: day 2—from 8 mmHg to 10 mmHg, day 6—from 5 mmHg to 6 mmHg, day 12—from 6 mmHg to 7 mmHg, day 16—from 5 mmHg to 7 mmHg, day 19—from 2 mmHg to 5 mmHg and, at day 21, at the time of definitive wall closure, the pressure increased from 2 mmHg to 7 mmHg. 

## 4. Discussion

Despite the surgical goal of reducing viscera in the abdomen in congenital abdominal wall defects, it is important to be cautious when selecting the type of abdominal wall closure. Overenthusiastic primary closure could lead, as we observed in our retrospective study, to abdominal compartment syndrome.

A 30-year retrospective study from a different Romanian hospital, regarding risk factors of the unfavorable evolution of their patients with gastroschisis, mentioned the presence of clinical signs of abdominal compartment syndrome in 19 out of 159 patients, reflecting percentages similar to ours [19]. These results reveal the fact that surgical experience is not enough to achieve the best results; we also need an objective quantification of abdominal pressure when performing abdominal closure.

Based on our results regarding the presence of postoperative abdominal compartment syndrome, we inferred that we should estimate the IAP at the time of abdominal wall closure as a way to improve outcomes for patients with abdominal wall defects. In this regard, we decided to implement, starting from June 2021, the protocol for intraabdominal pressure monitoring at the time of surgical procedure and performed primary closure only if the IVP was below 10 mmHg after integrating the viscera into the abdominal cavity.

Studying published articles regarding IVP measurement, we decided to measure the introduction of 1 mL/kg of sterile saline solution in the bladder. Defontaine et al. studied the required volume of saline solution instillation for IVP measurement and concluded that 1 mL/kg is the optimal volume for newborns weighing less than 4.5 kg [20]. Another study recorded IAP via an intraperitoneal dialysis catheter and an indwelling urinary catheter with different bladder filling volumes and determined that IAP is best estimated with an intravesical volume of 1 mL/kg [16].

Regarding the temperature of saline solution used for IVP measurement, a comparative study theorized that 35 °C saline solution, being close to body temperature, would not stimulate the bladder muscle wall. They compared the results of IVP using saline solution at 15 °C, 25 °C, and 35 °C. There were significant differences between IVP using 35 °C and 15 °C saline solution, probably because of bladder wall muscle stimulation due to lower temperatures. When comparing IVP values using 35 °C and 25 °C saline solution, the results were similar. They concluded that it is not necessary to heat the saline solution and that room-temperature solution could be directly used to measure IVP [21]. In consideration of this study, we agreed to use room-temperature solution for our measurements.

Nevertheless, the pressure should be measured with the patient in supine position with no elevation of the head of the table, because studies show increased values of IVP with bed elevation, probably due to gravitational visceral compression of the bladder [22,23,24]. 

For the patients included in the prospective study, clinical signs of abdominal compartment syndrome were absent and the IVP values monitored were all below 10 mmHg. In the cases where we performed staged closure, there were no complications related to the type of prosthetic material we used to create the bag for gradual reduction of the viscera.

Following our few cases after we implemented measurement of IVP, we are optimistic about the future regarding postoperative management of patients with congenital abdominal wall defects. Undoubtedly, we need a substantial number of cases in order to make a valid conclusion.

Unfortunately, as shown by a few studies, pediatric guidelines for IAH and ACS are not very widely acknowledged. In 2010, 127 heads of pediatric intensive care units from Germany responded to a questionnaire that revealed that the vast majority (99 respondents) use exclusively clinical signs for the diagnosis of IAH and ACS. Also, only 3.9% chose the correct IAH/ACS criteria and 16.5% chose definitions in accordance with WSACS consensus [25]. After the 2013 WSACS updated guidelines, 156 questionnaires were returned from heads of pediatric/neonatal intensive care units from Germany, Austria, and Switzerland. Only 6% of the physicians chose the correct definition for IAH and 58% chose the correct definition of ACS in conformity with WSACS updated guidelines. Furthermore, there were still a high number of centers that relied exclusively on clinical signs to diagnose IAH (50%) and ACS (40%) [26]. Other recent surveys from Saudi Arabia also revealed that many physicians are not aware of the actual definitions of IAH [27,28].

Presenting our experience aims to emphasize the importance of staying current in our field in order to improve our patients’ outcomes. Better results are linked to understanding our past results, observing what we could improve and, consequently, searching for solutions. Also, by enhanced cooperation between the surgical team and the neonatal intensive care team, we could find inventive solutions for our problems—for instance, in this case, we used for IVP measurement the pressure transducers that were already available in the Neonatal Intensive Care Unit for invasive blood pressure monitoring.

As a future perspective, we trust that, if we continue to perform abdominal closure only at pressures lower than 10 mmHg and monitor the IVP until we reach a plateau, there will be no postoperative compartment syndrome attributable to abdominal closure. In this manner, intravesical pressure measurement is of great value in preventing complications related to increased abdominal pressure, as it is a non-invasive method that could be easily applied either in the Operating Room or in the Neonatal Intensive Care Unit.

## 5. Conclusions

Developing abdominal compartment syndrome after the closure of congenital abdominal wall defects represents a risk even in the most experienced hands. For this reason, we believe that we should use a quantifiable method with which to assess intraabdominal pressure. As stated by the WSASC consensus, indirect measurement of intraabdominal pressure via a bladder catheter connected to a pressure transducer is a reliable method. In this manner, measuring the abdominal pressure while closing the abdominal wall and not allowing pressures above 10 mmHg would help us prevent postoperative abdominal compartment syndrome. This approach should aid us to achieve better postoperative evolution for patients with gastroschisis and omphalocele.

## Figures and Tables

**Figure 1 children-10-01384-f001:**
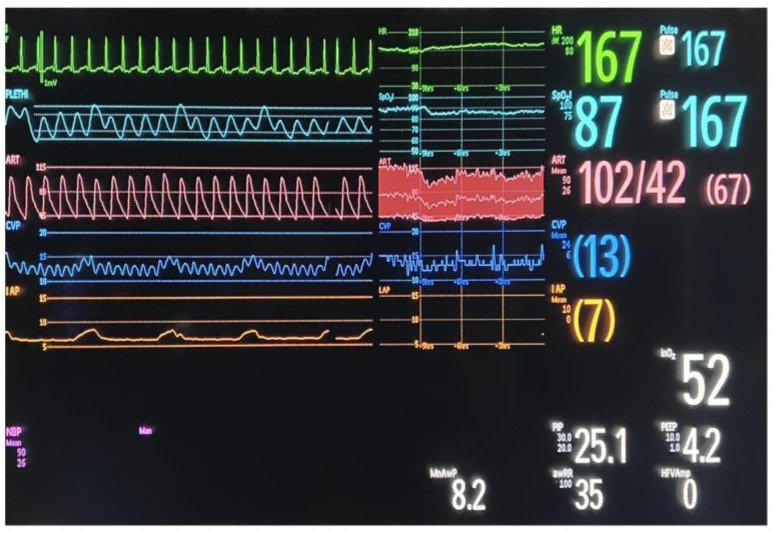
Value of IVP expressed in mmHg shown on the monitor.

**Figure 2 children-10-01384-f002:**
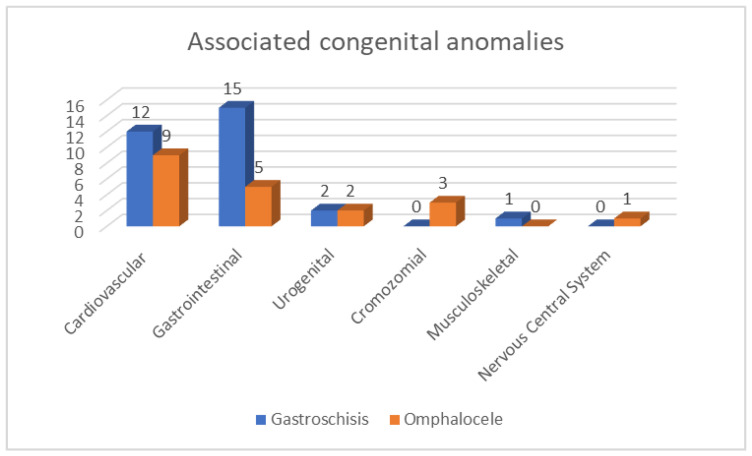
Distribution of associated congenital anomalies divided by the type of the abdominal wall defect.

**Figure 3 children-10-01384-f003:**
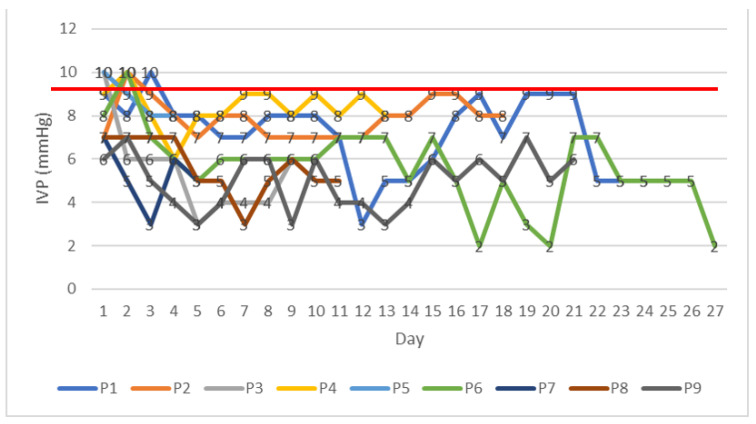
Daily values of IVP (mmHg) for each of the patients included in the study.

**Table 1 children-10-01384-t001:** Demographics of gastroschisis (G) versus omphalocele (O).

	**Birth Weight (g)**		**Gestational Age (Weeks)**		**Apgar Score**	
	**G**	**O**	**G**	**O**	**G**	**O**
Mean	2404	2681	36	36	7	8
Std. Deviation	508	592	2	2	1	1
Minimum	1260	1730	31	33	2	4
Maximum	3300	4200	40	41	9	9
**Independent Samples *t*-Test**						
				* **t** *	**df**	* **p** *
Birth Weight (g)				−1.691	45	0.098
Gestational age (weeks)				−0.370	45	0.713
Apgar Score				−1.578	45	0.122

Note. Student’s *t*-test.

**Table 2 children-10-01384-t002:** Surgical procedures performed for gastroschisis.

**Surgical Procedure for Abdominal Wall Closure**	**Frequency**	**Percent**
primary closure	27	90.0
staged closure using polyvinyl chloride bag	1	3.3
staged closure using silo bag	1	3.3
staged closure using polyvinyl chloride patch	1	3.3
**Additional procedures**	**Frequency**	**Percent**
abdominal wall stretching	5	16.7
skin flap dissection	1	3.3
intestinal resection and anastomosis	2	6.7
no additional procedure	16	53.3
umbilical cord patch	6	20.0

**Table 3 children-10-01384-t003:** Surgical procedures performed for omphalocele.

**Surgical Procedure for Abdominal Wall Closure**	**Frequency**	**Percent**
delayed closure—wound dressing for 28 days	1	5.9
primary closure	16	94.1
**Additional procedures**	**Frequency**	**Percent**
abdominal wall stretching	1	5.9
intestinal resection and Mikulicz ileostomy + lysis of Ladd’s peritoneal bands	1	5.9
no additional procedure	14	82.3
resection of omphaloenteric duct	1	5.9

**Table 4 children-10-01384-t004:** Time from birth until surgical procedure.

	Hours from Birth to Surgery	
	Gastroschisis	Omphalocele
Median	6	48
Mean	8	88
Std. Deviation	6	155
IQR	4	48
Minimum	3	7
Maximum	29	672

**Table 5 children-10-01384-t005:** Development of abdominal compartment syndrome related to the type of surgical procedure used for abdominal wall closure.

Surgical Procedure	Compartment Syndrome	Frequency	Percent
delayed closure (wound dressing)	No	1	100
	Yes	0	0
primary closure	No	38	88
	Yes	5	12
staged closure (polyvinyl chloride bag)	No	1	100
	Yes	0	0
staged closure (silo bag)	No	1	100
	Yes	0	0
staged closure (polyvinyl chloride patch)	No	0	0
	Yes	1	100

**Table 6 children-10-01384-t006:** Demographic data of patients included in the prospective study.

	Birth Weight (g)	Gestational Age (Weeks)	Apgar Score
Mean	2835	38	8
Std. Deviation	608	1	1
Minimum	2000	37	7
Maximum	3900	40	9

**Table 7 children-10-01384-t007:** Surgical procedures performed for gastroschisis and omphalocele.

**Diagnosis**	**Surgical Procedure**	**Frequency**
Gastroschisis	Primary closure	3
	Staged closure—Schuster procedure with PVC tailored bag	2
Omphalocele	Delayed closure—partial reduction and ligation of the omphalocele sac, next day abdominal wall closure	1
	Primary closure	1
	Staged closure—Schuster procedure with PVC tailored bag	2

**Table 8 children-10-01384-t008:** Diuresis values (mL/kg/h) for each of the patients included in the study. (Px = patient number x).

	P1	P2	P3	P4	P5	P6	P7	P8	P9
Mean	5.7	4.5	7.3	5.0	4.1	4.6	3.3	6.9	5.4
Std. Deviation	1.8	1.6	2.4	1.8	0.4	1.4	0.3	1.6	2.1
Minimum	2.0	1.0	3.5	2.5	3.7	2.6	3.1	5.4	2.6
Maximum	10.9	8.0	10.0	8.5	4.5	8.3	3.5	9.2	9.3

**Table 9 children-10-01384-t009:** Heart rate (beats per minute) for each of the patients included in the study.

	P1	P2	P3	P4	P5	P6	P7	P8	P9
Mean	165	141	156	150	125	134	144	123	157
Std. Deviation	12	24	13	17	6	12	33	18	15
Minimum	124	94	137	121	118	111	120	87	130
Maximum	182	181	175	185	130	162	167	140	198

**Table 10 children-10-01384-t010:** Mean Arterial Pressure (mmHg) for each of the patients included in the study.

	P1	P2	P3	P4	P5	P6	P7	P8	P9
Mean	51	59	52	56	44	52	51	63	54
Std. Deviation	8	8	8	5	8	6	5	6	7
Minimum	42	42	42	47	38	42	48	55	45
Maximum	66	71	66	63	53	63	58	72	71

**Table 11 children-10-01384-t011:** IVP values (mmHg) for each of the patients included in the study.

	P1	P2	P3	P4	P5	P6	P7	P8	P9
Mean	7	8	5	8	9	6	5	6	5
Std. Deviation	2	1	2	1	1	2	1	1	1
Minimum	3	7	3	6	8	2	3	3	3
Maximum	10	10	10	10	10	10	7	7	7

## Data Availability

Data sharing not applicable.

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
