# Peer review of "Avoiding High Pressure Abdominal Closure of Congenital Abdominal Wall Defects—One Step Further to Improve Outcomes"

_children, 2023, doi:10.3390/children10081384_

Round 1

Reviewer 1 Report

The introduction is excessively long. It is lost by explaining the measurement of intravesical pressure, when it is something that is routinely performed in these neonatal patients. The objective of the study is not clear. The last sentence of the introduction should be a clear and concise definition of the aim of the study, to situate the reader before starting to read the Methods used to achieve that aim.

The methods are very brief, without specifying the protocol followed in your centre for patients with gastroschisis and omphalocele. They do not explain at what age they intervene, nor the therapeutic possibilities that are offered or have been offered in the last 10 years in their centre.

The authors state that "Informed consent was obtained from the parents of all the subjects involved in the study". This is questionable given that this is a 10-year retrospective study. When was this consent obtained? At the time of the intervention? Was consent sought to operate on the patients or to publish the results of this study, which was decided 10 years after the first interventions on these children? 

Authors should take care of these aspects, because they affect the quality of the publications and the prestige of the journals.

In the results, the authors talk about whether or not there are differences between the gastroschisis and omphalocele groups without providing any statistical data to support these claims. No statistical analysis with SPSS or similar software? The statistical methodology leaves much to be desired. They do not perform multivariate analysis to determine the factors that influence the development of compartment syndrome. These results are not presentable, there is a lack of correct methodology and expression of the results.

For all these reasons, I do not consider that this article meets the minimum quality required for publication in this journal.

Moderate editing of English language required

Author Response

 Dear Editor of Children and Reviewers, 

We would like to thank you for the improvement suggestions and for the opportunity to resubmit a revised version of our manuscript children-2476276, entitled: ”Avoiding high pressure abdominal closure of congenital abdominal wall defects – one step further to improve outcomes”. 

We have carefully considered the reviewers’ comments and suggestions and we uploaded a revised manuscript. Revisions in the manuscript are shown in red font. 

We would like to address, point by point, the concerns raised by the reviewers. 

Reviewer 1: 

  • “The introduction is excessively long. It is lost by explaining the measurement of intravesical pressure, when it is something that is routinely performed in these neonatal patients.” 

We took into consideration this suggestion, and we have shortened the introduction. Since, in our hospital, measurement of intravesical pressure was not performed before the beginning of our prospective study in 2021, we considered useful to explain the manner we performed intravesical pressure measurement based on the recent literature. 

  • “The objective of the study is not clear. The last sentence of the introduction should be a clear and concise definition of the aim of the study, to situate the reader before starting to read the Methods used to achieve that aim.” 

This is a good point that has led to rewrite the last paragraph of introduction. 

  • “The methods are very brief, without specifying the protocol followed in your centre for patients with gastroschisis and omphalocele. They do not explain at what age they intervene, nor the therapeutic possibilities that are offered or have been offered in the last 10 years in their centre”. 

As of this suggestion, we included additional details regarding the time we perform surgery for patients with gastroschisis. These clarifications were introduced in the results section, where we present the range of hours from birth until surgical procedure - Table 4. 

  • “The authors state that "Informed consent was obtained from the parents of all the subjects involved in the study". This is questionable given that this is a 10-year retrospective study. When was this consent obtained? At the time of the intervention? Was consent sought to operate on the patients or to publish the results of this study, which was decided 10 years after the first interventions on these children?” 

Taking into account this message, we completed the information regarding informed consent. Until 2018, our informed consent consisted only on permission, concerning surgery and medical treatment. Since 2018, informed consent includes a section for data processing, for research studies and medical teaching, owing to the fact that our institution is a teaching hospital. 

For the patients admitted prior to 2018, data collection and results publishing were approved by the ethic committee of our hospital. 

  • “In the results, the authors talk about whether or not there are differences between the gastroschisis and omphalocele groups without providing any statistical data to support these claims. No statistical analysis with SPSS or similar software? The statistical methodology leaves much to be desired. They do not perform multivariate analysis to determine the factors that influence the development of compartment syndrome. These results are not presentable, there is a lack of correct methodology and expression of the results.” 

We modified Table 1, where we presented the descriptive statistics for demographics, and added the result of the Independent Sample T Test, performed with JASP statistical software, that showed no statistically significant difference between the omphalocele group and the gastroschisis group. 

Reviewer 2 Report

The development of abdominal compartment syndrome in children after closure of gastroschisis or omphalocele is an important clinical problem. 

In this retrospective study the authors found a frequency on 13% for developing symptoms of compartment syndrome in 47 patients over an inclusion period of 10 years. This is a relative high frequency and there is no information on the degree of compartment syndrome that was observed. There is no information on the diagnostic criteria for abdominal compartment syndrome used in the study. The symptoms may vary from weak and unspecific symptoms to overt multiorgan failure. Also, there is no information on the treatment of the compartment syndrome, which is a very important issue in relation to the seriousness of the syndrome in the different cases. This information is mandatory to understand the significance of this clinical problem.

Was surprised by the higher incidence of other congenital abnormalities in the group of patis with gastroschisis compared to omphalocele. It used to be the other way round?

The results of the prospective study must be included in the result resection, and it may then be deliberately commented in the Discussion section.

The extensive number of decimals should be decreased to one or none both in the text and in the table. 

The introduction can be shortened.  

Author Response

 Dear Editor of Children and Reviewers, 

We would like to thank you for the improvement suggestions and for the opportunity to resubmit a revised version of our manuscript children-2476276, entitled: ”Avoiding high pressure abdominal closure of congenital abdominal wall defects – one step further to improve outcomes”. 

We have carefully considered the reviewers’ comments and suggestions and we uploaded a revised manuscript. Revisions in the manuscript are shown in red font. 

We would like to address, point by point, the concerns raised by the reviewers. 

  • “In this retrospective study the authors found a frequency on 13% for deing symptoms of compartment syndrome in 47 patients over an inclusion period of 10 years. This is a relative high frequency and there is no information on the degree of compartment syndrome that was observed. There is no information on the diagnostic criteria for abdominal compartment syndrome used in the study. The symptoms may vary from weak and unspecific symptoms to overt multiorgan failure. Also, there is no information on the treatment of the compartment syndrome, which is a very important issue in relation to the seriousness of the syndrome in the different cases. This information is mandatory to understand the significance of this clinical problem.” 

We acknowledged this suggestion and we included, in the results section, additional information regarding the diagnosis and management of patients with abdominal compartment syndrome included in our study. 

  • “Was surprised by the higher incidence of other congenital abnormalities in the group of patients with gastroschisis compared to omphalocele. It used to be the other way round?” 

We noticed that our data regarding additional congenital malformations do not correspond with the ones in the literature. We reexamined our database but we were not able to find a different result. Perhaps we should perform another retrospective study focused on associated anomalies to be able to explain these results. 

  • “The results of the prospective study must be included in the result resection, and it may then be deliberately commented in the Discussion section” 

We took into consideration this suggestion and we included the prospective study in the results section introducing additional result of our ongoing prospective study. 

  • “The extensive number of decimals should be decreased to one or none both in the text and in the table.” 

We decreased the decimals according to this suggestion. 

  • “The introduction can be shortened” 

As suggested by both reviewers, we shortened the introduction. 

Thank you, again, for your consideration of our revised manuscript. 

Sincerely, 

Catalin Cirstoveanu, MD, PhD 

Department of Neonatal Care Unit “Marie Curie” Emergency Children’s Hospital, Bucharest, Romania 

Round 2

Reviewer 1 Report

The authors have responded appropriately to the recommended suggestions.

The manuscript is ready for publication.

Moderate editing of English language required

Author Response

Dear Editor of Children and Reviewers,

We would like to express our gratitude to the editor and the reviewers for the time they invested in our article.

Please find below our replies to the second round of reviews for our manuscript children-2476276 entitled „Avoiding high pressure abdominal closure of congenital abdominal wall defects – one step further to improve outcomes”. New revisions in the manuscript are highlighted in yellow.

  • Reviewer 1

„Comments and Suggestions for Authors:

The authors have responded appropriately to the recommended suggestions.

The manuscript is ready for publication.

Comments on the Quality of English Language

Moderate editing of English language required.”

We checked again the spelling and we tried to improve some of the expressions.

Reviewer 2 Report

The revised manuscript has improved considerably. The clinical delineation of compartment syndrome is difficult. Four of the six patients that developed compartment syndrome were operated du to ileus caused by either stenosis or adhesions. Two patients were treated medically. Were the 4 patients operated because of an ileus diagnosed on abdominal plain X-ray? In true abdominal compartment syndrome, the abdomen is left open! Whys was is closed in 3 of the 4 patients? 

Author Response

Dear Editor of Children and Reviewers,

We would like to express our gratitude to the editor and the reviewers for the time they invested in our article.

Please find below our replies to the second round of reviews for our manuscript children-2476276 entitled „Avoiding high pressure abdominal closure of congenital abdominal wall defects – one step further to improve outcomes”. New revisions in the manuscript are highlighted in yellow.

The revised manuscript has improved considerably. The clinical delineation of compartment syndrome is difficult. Four of the six patients that developed compartment syndrome were operated du to ileus caused by either stenosis or adhesions. Two patients were treated medically. Were the 4 patients operated because of an ileus diagnosed on abdominal plain X-ray? In true abdominal compartment syndrome, the abdomen is left open! Whys was is closed in 3 of the 4 patients?

We agree that compartment syndrome is difficult to diagnose without being able to have an objective measurement of intraabdominal pressure.

Intestinal obstruction was diagnosed based on clinical signs and on abdominal plain X-ray.

During the surgical procedure, after lysis of the adhesion in two cases and segmental jejunal resection in one case, gentle maneuvers for intestinal content evacuation were performed. In that manner, the previously distended bowel loops were easily reintegrated in the abdominal cavity and closure of the abdominal without tension was possible. In one case, after resection of the stenotic segment and ileostomy, primary closure was not possible due to abdominal-visceral disproportion. In that case, temporary abdominall wall closure was achieved using prosthetic material. Taking into consideration the reviewer’s question, we included in the manuscript justification for the decision to perform primary closure of the abdominal wall or temporary closure with prosthetic material.

Thank you again for your consideration of our revised manuscript.

Sincerely,

Catalin Cirstoveanu, MD, PhD

Department of Neonatal Care Unit “Marie Curie” Emergency Children’s Hospital, Bucharest, Romania
